# O-GlcNAcylation: Crosstalk between Hemostasis, Inflammation, and Cancer

**DOI:** 10.3390/ijms25189896

**Published:** 2024-09-13

**Authors:** Itzel Patricia Vásquez Martínez, Eduardo Pérez-Campos, Laura Pérez-Campos Mayoral, Holanda Isabel Cruz Luis, María del Socorro Pina Canseco, Edgar Zenteno, Irma Leticia Bazán Salinas, Margarito Martínez Cruz, Eduardo Pérez-Campos Mayoral, María Teresa Hernández-Huerta

**Affiliations:** 1UNAM-UABJO Faculty of Medicine Research Center, Faculty of Medicine and Surgery, Autonomous University “Benito Juarez” of Oaxaca, Oaxaca 68020, Mexico; vami940508.fmc@uabjo.mx (I.P.V.M.); lperez.cat@uabjo.mx (L.P.-C.M.); holanda.crz@gmail.com (H.I.C.L.); mpina.cat@uabjo.mx (M.d.S.P.C.); letitabs@hotmail.com (I.L.B.S.); eperezcampos.fmc@uabjo.mx (E.P.-C.M.); 2National Institute of Technology of Mexico, Technological Institute of Oaxaca, Oaxaca 68033, Mexico; pcampos@itoaxaca.edu.mx (E.P.-C.); mcruz@itoaxaca.edu.mx (M.M.C.); 3Department of Biochemistry, Faculty of Medicine, National Autonomous University of Mexico, Mexico City 04510, Mexico; ezenteno@unam.mx; 4National Council of Humanities, Sciences and Technologies (CONAHCYT), Faculty of Medicine and Surgery, Autonomous University “Benito Juarez” of Oaxaca, Oaxaca 68120, Mexico

**Keywords:** O-GlcNAc, hemostasis, inflammation, cancer, post-translational modification

## Abstract

O-linked β-N-acetylglucosamine (O-GlcNAc, O-GlcNAcylation) is a post-translational modification of serine/threonine residues of proteins. Alterations in O-GlcNAcylation have been implicated in several types of cancer, regulation of tumor progression, inflammation, and thrombosis through its interaction with signaling pathways. We aim to explore the relationship between O-GlcNAcylation and hemostasis, inflammation, and cancer, which could serve as potential prognostic tools or clinical predictions for cancer patients’ healthcare and as an approach to combat cancer. We found that cancer is characterized by high glucose demand and consumption, a chronic inflammatory state, a state of hypercoagulability, and platelet hyperaggregability that favors thrombosis; the latter is a major cause of death in these patients. Furthermore, we review transcription factors and pathways associated with O-GlcNAcylation, thrombosis, inflammation, and cancer, such as the PI3K/Akt/c-Myc pathway, the nuclear factor kappa B pathway, and the PI3K/AKT/mTOR pathway. We also review infectious agents associated with cancer and chronic inflammation and potential inhibitors of cancer cell development. We conclude that it is necessary to approach both the diagnosis and treatment of cancer as a network in which multiple signaling pathways are integrated, and to search for a combination of potential drugs that regulate this signaling network.

## 1. Introduction

Post-translational protein modifications (PTM) allow cells to respond to cellular or environmental signals. O-linked β-d-*N*-acetylglucosamine (O-GlcNAc, O-GlcNAcylation, or O-acetylglucosaminylation) is a highly dynamic and reversible PTM process [1,2], through the addition of N-acetylglucosamine (GlcNAc) to the hydroxyl group of serine or threonine residues into nuclear, cytoplasmic, and mitochondrial proteins [3]. The enzymes responsible for O-GlcNAcylation are O-GlcNAc transferase (OGT) and O-GlcNAcase (OGA), which are responsible for adding and removing GlcNAc to proteins, respectively. O-GlcNAcylation is the endpoint of the hexosamine biosynthetic pathway (HBP) and terminates with uridine diphosphate N-acetylglucosamine (UDP-GlcNAc), which is the substrate of OGT [4,5]. UDP-GlcNAc biosynthesis is regulated by almost all metabolic pathways in the cell [2] (Figure 1).

O-GlcNAcylation participates in many cellular processes [6] such as signal transduction [7], transcription [8], cell cycle control [9,10], epigenetic control of gene expressions in response to stress [11], and nutrient imbalance [12]. It is also present in almost all cellular compartments; and in humans, it is more abundant in the pancreas, liver, brain, skeletal muscle, adipose tissue, and other organs and tissues [13]. For this reason, O-GlcNAcylation alterations have been related to metabolic diseases [14,15], cardiovascular diseases [16], neurodegenerative diseases [17,18], autoimmune diseases, and cancer [19] (Figure 2).

Furthermore, O-GlcNAcylation alters signaling pathways, particularly in cancer, by promoting tumor growth and immune escape. Among the signaling pathways that are modified are the HBP, which is associated with excess glucose and cancer [20], and the O-GlcNAc/c-Myc pathway, which is involved in the regulation of megakaryopoiesis and thrombopoiesis and, therefore, in hemostasis [21]. Platelets can modulate tumor-associated inflammation and stimulate metastasis and venous thrombosis. Tumor cell-induced platelet aggregation has been demonstrated in several cell lines. In addition, other important molecules in the signaling of thrombo-inflammation and cancer are thrombin, tumor-expressed ADP, podoplanin (PDPN), tissue factor (TF), and ADAM-13, among many others [22]. The most studied pathways associated with O-GlcNAcylation are thrombosis, inflammation and cancer in the PI3K/Akt/c-Myc pathway [23], the nuclear factor-kappa B (NF-κB) signaling pathway [24,25], and PI3K/AKT/mTOR pathway [26,27] (Figure 3, modified of https://www.kegg.jp/pathway/map=map05200&keyword=cancer (accessed on 2 September 2024)). In recent years, the role of O-GlcNacylation has been demonstrated in cancer cell proliferation, angiogenesis, and metastasis, as well as in the state of cancer-associated inflammation, and drug resistance has been demonstrated. In this review, we aim to explore the relationship between O-GlcNAcylation and hemostasis, inflammation, and cancer, which could serve as potential prognostic tools or clinical predictions for cancer patients’ healthcare and as an approach to combat cancer.

## 2. Cancer, Hemostasis, and O-GlcNAcylation

Cancer-associated thrombosis is a major cause of morbidity and mortality for cancer patients [28]. Likewise, tumors can induce platelet activation, aggregation, release of platelet-derived macrovesicles into circulation, and promote thrombocytosis (Figure 2). There are factors associated with cancer and thrombosis that have been recently reviewed, such as leukocytes (eosinophils, monocytes, and neutrophils), TF, thrombocytosis, and its leukocyte-related indices [29], PDPN, plasminogen activator inhibitor-1, the intrinsic coagulation pathway, von Willebrand factor [30], and miRNAs (e.g., miR-126) [31]. In this way, increased platelet count and platelet-associated clinical laboratory indexes could be used as predictive biomarkers of tumors [29]. In cancer cells, O-GlcNAc and its enzyme OGT are highly elevated, thus coupling the alteration in nutrient status to signaling activities, contributing to reprogramming cellular metabolism and cancer progression [32]. Blood O-GlcNAc levels have been described as increasing in response to inflammation and stress tissue [33].

Cancer cells reprogram their energy metabolism and signaling networks to promote growth, survival, proliferation, and long-term maintenance. Cancer progression is characterized by an altered metabolic state and even different metabolic phenotypes in the cells of the same cancer. Under normal conditions, cells process glucose under aerobic conditions and promote glycolysis under anaerobic conditions; here, the final product of glycolysis depends on the level of available oxygen (Pasteur effect). However, even in high oxygen, cancer cells prefer glycolysis instead of oxidative phosphorylation, called the Warburg effect [26]. This metabolic reprogramming allows cancer cells to absorb a large part of the nutrients from their environment (glucose and glutamine) producing low amounts of adenosine triphosphate (ATP) as a by-product and secreting large amounts of carbon and nitrogen in the form of lactate and ammonia into the extracellular environment so that they can be used by tumor cells as fuel [34,35]. In addition, cancer and proliferating cells adjust their energy metabolism by increasing glucose concentration when there is a decrease in oxidative phosphorylation via the Crabtree effect [36]. In summary, cancer cells are characterized by a high demand and consumption of glucose.

Excess glucose in cancer cells primarily enters glycolysis and increases flow to branching glucose pathways such as HBP [37]. HBP integrates pathways of glucose, amino acid, fatty acid, and nucleotide metabolism to promote the synthesis of UDP-GlcNAc, the end product of HBP and substrate for O-GlcNAcylation [38]. Activation of HBP promotes the proliferation of lung cancer cells, and the inhibition of enzymes related to HBP such as glutamine fructose-6-phosphate amidotransferase 1 (GFAT1) reduces programmed death ligand 1 (PD-L1) levels and, therefore, the progression of lung cancer. O-GlcNAc suppresses the degradation of PD-L1 in cancer cells, which is responsible for immune evasion in cancer [39]; high expression of PD-L1 in tumor cells has been correlated with poor prognosis in cancer patients, such as non-small cell lung cancer (NSCLC) [40]. While studies show that PD-L1 does not appear to be associated with the incidence of venous thromboembolism in NSCLC [41], others have found that the presence of PD-L1 in NSCLC may be associated with an increased risk of thromboembolism [42].

Oncogenes, tumor suppressors, and proteins involved in tumor biology are also regulated by O-GlcNAc [43] such as phosphoglycerate kinase 1 (PGK1), enolase 1 (ENO1), glucose-6-phosphate dehydrogenase (G6PD) c-Myc, p53, Ras, AMPK, and NF-κB, among others. PGK1 regulates the expression of urokinase-type plasminogen activator receptor (uPAR) mRNA associated with metastasis in gastric cancer and breast cancer [44,45]. O-GlcNAcylation activates PGK1 activity to enhance lactate production and simultaneously induces PGK1 translocation into mitochondria to inhibit the pyruvate dehydrogenase (PDH) complex and, thus, reduce oxidative phosphorylation [46]. Excessive lactate production by tumor and stromal cells is associated with increased aggressiveness, due to extracellular acidification, which also induces invasion and metastasis, inhibition of the antitumor immune response, and resistance to therapy [47]. Increased blood lactate levels impair the coagulation system [48].

ENO1 acts as a plasminogen receptor that is converted into plasmin and induces fibrinolysis; in cancer cells, it enhances their ability to invade through the stroma, mainly by activating collagenases and degrading fibrin [49]. ENO1 is associated with a better prognosis only in early-stage breast cancer, which may be related to the different effects of ENO1 on immune infiltration [50]. ENO1 promotes glycolysis, cell proliferation, and migration through activation of the phosphatidylinositol 3-kinase (PI3K)/Akt and adenosine monophosphate-activated protein kinase (AMPK)/mTOR pathways [51,52,53,54], and c-Myc pathways [55].

G6PD (EC 1.1.1.49) is responsible of the production of nicotinamide adenine dinucleotide phosphate (NADPH) for the prevention of cellular damage caused by reactive oxygen species. In response to increased oxidative stress, a reduced ability to induce the innate immune response has been found in G6PD-deficient cells [56]. O-GlcNAcylation of G6PD facilitates lung cancer while phosphorylation of G6PD by polo-like kinase 1 (Plk1) affects cell cycle progression and cell proliferation of multiple cancers [57]. Furthermore, primary G6PD deficiency can cause non-immune hemolytic anemia, whereas severe deficiency presents with intravascular hemolysis and, therefore, renal failure [58]. Other studies suggest that G6PD deficiency is associated with increased release of the proinflammatory cytokine IL-8, and decreased release of the anti-inflammatory cytokine IL-10 [59]. This is important because IL-8-induced O-GlcNAc modification through glucose transporter 3 (GLUT3) and glucosamine fructose-6-phosphate aminotransferase (GFAT) has been reported to regulate cancer stem cell-like properties in colon and lung cancer cells [60].

c-Myc is a helix–loop–helix leucine transcription factor involved in many cellular processes, including cell potency, apoptosis, and differentiation [61]. It regulates genes involved in glycolysis, purine/pyrimidine metabolism, and lipids [62], as well as genes involved in glutamine metabolism, mitochondria biosynthesis, cell cycle control, and HBP genes. In cancer cells, c-Myc is often elevated, which promotes energy production and biomolecule synthesis [63] by regulating genes for anabolic enzymes such as aspartate transcarbamylase and dihydroorotase (CAD), serine hydroxymethyl transferase (SHMT), fatty acid synthase (FAS), and ornithine decarboxylase (ODC). Elevated c-Myc expression in cancer cells has been reported to increase the need for glutamine [64]. Increased O-GlcNAcylation is observed in cancers, such as breast, pancreatic, liver, lung, colon [65], and leukemia [6]. O-GlcNAcylation participates in initiation, progression, and metastasis through various pathways, such as O-GlcNAcylation of G6PD in the pentose phosphate pathway (PPP) [66], O-GlcNAcylation of NF-κB and p53 [10,67] and c-Myc. Increased O-GlcNAcylation of c-Myc promotes the proliferation of pre-B cells as well as some B-cell cancers, such as chronic lymphocytic leukemia [6] and B acute lymphocytic leukemia [68]. Furthermore, the glycolytic pathway is modulated by the PI3K/Akt/c-Myc pathway in pre-B acute lymphoblastic leukemia (pre-ALL) [65]. Interconnections of c-Myc through the O-GlcNAc/c-Myc axis confer on it a regulatory function of O-GlcNAcylation in the processes of megakaryopoiesis and thrombopoiesis [69]. In the case of colon cells, their proliferation is related to platelets, which induce the positive regulation of the c-Myc oncoprotein. This upregulation of platelets can be reduced with aspirin and is correlated with the negative regulation of COX-2 and c-Myc expression [70].

The PI3K/Akt/mTOR signaling pathway is a key mechanism involved in the growth and control of glucose metabolism in cells [71]. The PI3K/Akt pathway regulates the uptake and utilization of glucose [72], its activation increases the expression of glucose transporters on the cell surface, while the activation of hexokinase (HK) and the phosphofructokinase-2-dependent allosteric activation of phosphofructokinase-1 (PFK1) phosphorylate glucose to promote glycolysis [73]. Furthermore, activation of the PI3K/Akt/mTOR pathway enhances carbohydrate, lipid, and protein biosynthesis. PI3K and Akt promote carbon flux from glucose into mitochondrial-dependent biosynthetic pathways such as fatty acid, cholesterol, and isoprenoid synthesis [74], while mTOR regulates cell growth from various amino acid precursors from transamination of mitochondrial intermediates (Figure 4).

The PI3K/Akt/mTOR signaling pathway has been described as one of the most activated pathways in various types of cancer [75], as a result of mutations in the phosphatidylinositol-4,5-bisphosphate 3-kinase catalytic subunit alpha (PIK3CA) or by the loss of phosphatidylinositol-3,4,5-trisphosphate 3-phosphatase (PTEN). Activation of this pathway in human cancers is caused by somatic alterations at specific sites in the pathway and by activation by receptor tyrosine kinases (RTKs) [76]. Somatic mutations in PIK3CA are common in a variety of tumor types including breast, colon, endometrial, and glioblastoma [77,78].

In prostate cancer, common pathways between HBP and PI3K/AKT/mTOR are observed. These pathways promote the expression and levels of androgen receptor, and this increase is mediated, in part, by the overexpression of OGT and stability of the pro-oncogene Myc, which benefits proliferation and metastasis [79]. Patients with prostate cancer have an increased risk of developing thromboembolic disease, which is increased by hormonal treatment [80]. On the other hand, the Akt kinase of the PI3K/AKT/mTOR pathway is associated with platelet activation [81], which, together with mutations in PI3K enzymes, may explain both metastasis and thrombosis in prostate cancer [11].

In the Src family, some members have been linked to O-GlcNAcylation and cancer, such as p53 [82] and p53/56lyn [83]. Mutations in p53 are the most common genetic change in cancer [84]. p53 is a transcription factor and tumor suppressor known for its involvement in various signaling pathways such as cell cycle arrest, cell apoptosis, autophagy, metabolism, response to DNA damage, and apoptosis; it plays an important role in the regulation of glycolysis and oxidative phosphorylation [85,86]. p53 can also arrest the cell cycle to activate DNA repair pathways; if the damage is extensive and cannot be repaired, it will induce the transcription of proteins involved in cell death by apoptosis [87]. In addition, p53 decreases the glycolytic rate through various mechanisms, represses the transcription of glucose transporters (GLUT) [88], and the translocation of GLUT1 to the plasma membrane [89] to suppress glucose uptake. It downregulates the levels of enzymes such as HK [90] and protein phosphoglycerate mutase 1 (PGAM1) that inhibit glycolysis [91], Figure 5. p53 inhibits the PPP to suppress glucose consumption through its binding to G6PD [92], and it also negatively regulates PI3K/Akt signaling through PTEN transcriptional induction [93]. However, in cancer cells, p53 is suppressed, resulting in loss of control of its functions, promoting glycolysis [94].

p53/56lyn is involved in polymorphonuclear leukocyte (PMN) [95], platelets signaling [96], and in TGF-beta 1-induced apoptosis in M-07e leukemic cells [97]. TNF-alpha-mediated stimulation of human PMN adherent to fibrinogen involves p53/56lyn [92] and could be related to neutrophil defects and inflammation [98]. Furthermore, p53/56lyn, along with other Src family kinases, is involved in signaling through the collagen receptor GPVI on the platelet [93].

O-GlcNAcylation has been considered as a “nutritional sensor”, since it exerts effects on the regulation of cell signaling, and transcription in cancer and inflammatory cells in response to nutrients and stress in the tumor microenvironment (TME) [30]. The signaling pathways transform environmental signals into intracellular events, such as immune cell activation and inflammation [99]. In addition, it interacts with immune evasion mechanisms and signaling pathways involved in thromboregulation and thrombosis in cancer-associated [100], e.g., some studies indicate that clotting protease thrombin, FVIIa, and FXa contribute to cancer immune evasion via unique mechanisms such as TF/FVIIa/PAR2 signaling [101].

## 3. O-GlcNAcylation and Inflammatory State of Cancer

The relationship between inflammation and cancer was first discovered in 1893 by Rudolf Virchow, who observed leukocytes infiltrating tumor tissue, suggesting that cancer may arise from chronic inflammation [102]. Inflammation is defined as the body’s response to tissue damage caused by physical injury, ischemic injury, infection, exposure to toxins, or other types of traumas. The inflammatory response triggers cellular changes and immune responses that result in the repair of damaged tissue and cell proliferation at the site of the injured tissue [103].

O-GlcNAcylation promotes precancerous inflammation and acts as a key orchestrator at the intersection of intrinsic and extrinsic inflammation through O-GlcNAcylation of key transcription factors and functional proteins in the activation of inflammatory cells to trigger cancer-associated inflammation in the TME [29].

Cancer-extrinsic inflammation is characterized by long-term chronic inflammatory conditions that predispose a person to cancer [104]. Risk factors related to this type of inflammation include bacterial and viral infections, obesity, autoimmune diseases, smoking, and excessive alcohol consumption. Approximately 20% of all cancers are related to chronic infection, chronic inflammation, or autoimmunity, in the same tissue or organ (Table 1) by intervening in various signal pathways with promotion (proliferation, metastasis, stemness, or drug resistance) or inhibitory effects (apoptosis) [105].

Cancer-intrinsic inflammation develops in most cancers, where cancer cells recruit immune cells and secrete inflammatory mediators to remodel the TME and, thus, promote cancer progression. This type of inflammation is caused by genetic and/or epigenetic mutations (oncogenes) [100]. Therefore, both extrinsic and intrinsic inflammation trigger the activation of transcription factors and signaling pathways to regulate the inflammatory response through soluble mediators (cytokines, chemokines) and other cellular components [35].

During cancer inflammation, several signal transduction pathways are deregulated to stimulate malignant transformation. Transcription factors associated with O-GlcNacylation and cancer (Table 2) such as NF-κB, the family of activators of transcription (STAT) STAT1/STAT3, and hypoxia-inducible factor (HIF) modulate the inflammatory response through inflammatory mediators (cytokines, chemokines) and immune cell infiltration promoting tumorigenesis [35].

NF-κB is a family of transcription factors including RelA/p65, RelB, c-Rel/Rel, p105/p50, and p100/p52 [126], and plays an important role in inflammatory, immune, and anti-apoptotic responses and hemostasis [127,128,129]. NF-κB is implicated in atherosclerosis and its pathological complication in atherothrombotic diseases due to its transcriptional role in maintaining pro-survival and pro-inflammatory states in vascular and blood cells [125]; this has a role in platelet survival, priming, activation, and aggregation. O-GlcNAcylation of NF-κB is involved in hyperglycemia-induced NF-κB activation and is required for lymphocyte activation [130]. NF-κB O-GlcNAcylation by increased uptake of glucose and glutamine by cancer cells regulates cancer cell proliferation, survival, and metastasis and acts as a link between inflammation and cancer [10,131].

Together with NF-κB, the Janus activator of transcription and signal transduction/kinase (JAK–STAT) signaling pathway is involved in the regulation of cytokine-dependent inflammation and immunity in carcinogenesis [132]. The binding of various ligands to cell surface receptors causes the receptor to activate the JAK family, which phosphorylate tyrosine residues on the receptor and recruits STAT family proteins. The STAT family comprises seven members: STAT1, STAT2, STAT3, STAT4, STAT5A, STAT5B, and STAT6 [133]. Some studies indicate that O-GlcNAcylation may regulate cancer-related inflammation independently or by modulating the production of HIF proteins [134].
ijms-25-09896-t002_Table 2Table 2Transcription factors associated with O-GlcNacylation and cancer.Transcription FactorsO-GlcNAcylation SiteEffectsType of CancerReferencesNF-κB p65Thr322Thr352↑ IL-6, TNF-α. Better invasion and metastasis capacity.↑ CXCR4 expression promotes metastasis. Cervix[135,136]NF-κB c-RelSer350↑ IL2, IFN-γ, GM-CSF.↑ c-Rel Pancreatic[137,138]TAB1Ser395Regulator ↑NF-κB ↑ IL-6 and TNF-α.↑ TAB1 Non-small celllung carcinoma[139,140]TAB3Ser408↑ Metastasis, NF-κBBreast[141]STAT3Thr717↑ Migration and invasion by ↑ IL-6/STAT3 signaling.Lung[142,143]STAT5AThr92↑ Oncogenic transcription, myeloid transformation. Leukemic[144]RIPK3Thr467↓ Inflammation, inflammation-associated necroptosis.↑ mRNA RIPk3 Hepatocellular, cervix[145,146,147]Abbreviations: NF-κB: nuclear factor κB; STAT: activator of transcription; TAB: activated binding adapter kinase 1 (TAK1) proteins; CXCR4: C-X-C motif chemokine receptor 4; TNF-α: tumor necrosis factor α; IFN-γ: interferon γ; GM-CSF: granulocyte–macrophage colony-stimulating factor; IL: interleukine; mRNA: messenger ribonucleic acid; ↑ increase; ↓ decreases.


## 4. O-GlcNAcylation, Inflammation, Hypercoagulation, and Cancer

In 1856, Rudolf Virchow postulated a triad of conditions that lead to thrombosis: endothelial injury, circulatory stasis, and abnormalities in the components of blood coagulation (hypercoagulable state, hypercoagulation) [148]. A hypercoagulable state occurs when the activation of the hemostatic mechanism of the plasma exceeds its physiological anticoagulant capacity, resulting in a predominance of prothrombotic activities [149].

The cancer-induced hypercoagulable state contributes to thrombosis, which is one of the main causes of death in cancer [150]. In cancer patients, hypercoagulability has been associated with an increased risk of venous thromboembolism, as well as with proliferation, tumor progression, and metastatic spread [151,152].

Different pathophysiological pathways have been identified regarding the interaction between cancer and the different components of the hemostatic system. Overexpression of procoagulant molecules such as TF has been found in various types of cancer. Likewise, plasminogen activator inhibitor 1 (PAI-1) is overexpressed and, as the main inhibitor of fibrin degradation, contributes to the procoagulant state of the TME. The expression of substances with effects on hemostasis has been reported in different types of cancer, some of which are mentioned in Table 3.

O-GlcNAc regulates the expression of PAI-1, fibronectin, and transforming growth factor-β by high glucose [153]. Another mechanism associated with facilitating immune evasion in breast cancer is that mediated by TF-FVIIa through the induction of PD-L1 [154]. O-GlcNAcylation also facilitates immune evasion through inhibition of lysosomal degradation of PD-L1 [39].
ijms-25-09896-t003_Table 3Table 3Hemostatic abnormalities in cancer.Type of CancerType of PatientsAbnormalities in HemostasisModifications O-GlcNAcReferencesBreastPatients with metastatic breast cancer↑ D-dimer, ↑ fibrinogen, ↑ prothrombin fragment 1 + 2. There is a correlation between hyperactive/activated platelets and tumor progression and thrombus formation.O-GlcNAc modification regulates MTA1 transcriptional activity during breast cancer cell genotoxic adaptation.[125,148,155,156,157,158]ProstatePatients with prostate cancer tissues and enhances malignancy of prostate cancer cells↑ VWFA higher incidence for venous thromboembolisms in prostate cancer exists. Platelets synthesize testosterone in patients with prostate cancer, which may explain relapses in castration-resistant prostate cancer. Insulin resistance in prostate cancer patients predisposes them to acute ischemic heart disease through dermcidin, a protein that can induce abnormal platelet aggregation.Inhibiting the formation of the E-cadherin/catenin/cytoskeleton complex may underly the O-GlcNAc-induced prostate cancer progression.[146,159,160,161,162,163]OvaryPatients with ovarian cancer without treatment↑TFThrombophilia due to a factor V and prothrombin mutation.↑ D-dimer, hs-CRP, and IL-6 compared to the control group. Tissue factor pathway inhibitor (TFPI)-2 has been implicated in the suppression of epithelial ovarian cancer, particularly clear cell carcinoma (CCC). It negatively regulates plasmin. TFPI-2 is secreted by CCC, platelets, and vascular endothelial cells.O-GlcNAcylation increases the motility of ovarian cancer cells via the RhoA/ROCK/MLC signaling pathway.[164,165,166,167,168]CervixPatients with adenosquamous carcinoma, adenocarcinoma, or squamous cell carcinoma↑ TF. Pretreatment thrombocytosis predicts pelvic lymph node metastasis and larger tumor size. It is also associated with markers of subclinical inflammation such as the neutrophil to lymphocyte ratio (NLR) and the platelet to lymphocyte ratio (PLR).Increased O-GlcNAcylation promotes IGF-1 receptor/phosphatidyl inositol-3 kinase/Akt pathway.HPV E6 upregulates OGT, increases O-GlcNAc, stabilizes c-MYC via O-GlcNAc, and enhances HPV oncogene activities.[169,170,171,172,173]Colorectal1. Patients with worse prognosis.2. Patients with colorectal cancer diagnosed without treatment.3. Recently diagnosed patients1. ↑ VWF in patients with worse prognosis.↑ D-dimer and ↑ fibrinogen.2. ↑ P-Selectin, CRP, IL-6 compared to the control group, ↑ in the group with metastasis compared to the group without metastasis.3. ↑ IL-6, IL-1b, and TNF-α compared to the control group. ↑ IL-6, CRP, and fibrinogen are more advanced stages. Platelet infiltration in tumors, particularly in CRC, is associated with a poor prognosis, as observed in the analysis of postoperative survival.O-GlcNAcylation, which is negatively regulated by microRNA-101, likely promotes CRC metastasis by enhancing EZH2 protein stability and function.[174,175,176,177,178]PancreasPatients with pancreatic adenocarcinomas are at increased risk for hypercoagulability↑ TF in pancreatic neoplasms↑ TF ↑ VEGF in carcinomas is related to a higher rate of venous thromboembolism (26.3%)Preoperative hypercoagulability can be identified with rotational thromboelastometry and is associated with lymphovascular/perineural invasion and advanced-staged disease in cancer.Glycosylation of Sox2 by OGT can affect its transcriptional activity and thereby regulate self-renewal in cancer.[179,180,181]GastricPatients in early, advanced stages or metastasis↑ IL-6 in patients compared to the control group, ↑ in early-stage patients, ↑ number of platelets in the locally advanced and metastatic groups ↑ CRP in cancer patients, ↑ platelets, MPV, and LPLT or PLT in the metastasis group; in addition, MPV ↑ is associated with active inflammation.Hyper-O-GlcNAcylation significantly promotes GC cells proliferation by modulating cell cycle related proteins and ERK 1/2 signaling.O-GlcNAcylation on RTN2 was pivotal for its oncogenic functions in gastric cancer.[182,183,184,185]LungPatients with early lung cancer (stage I and II) treated with surgery↓ D-dimer, MLR, NLR, and PLR are associated with significantly better overall survival.Hyper-O-GlcNAcylation induces cisplatin resistance via regulation of p53 and c-Myc in human lung carcinoma.[62,186]HepatocellularPatients with early lung cancer (stage I and II) treated with surgery↑ D-dimer in AFP-negative patients.↑ Prealbumin was correlated with tumor size.HCC-related cerebral infarction patients are at high risk of hypercoagulabilityOGT mediated by O-GlcNAcylation stabilized RAB10, thus accelerating HCC progression.OGT plays an oncogenic role in NAFLD-associated HCC by regulating palmitic acid and inducing ER stress, activating oncogenic JNK/c-Jun/AP-1 and NF-κB cascades.[187,188,189,190]Leukemia and lymphomaConsecutive patients with acute leukemia and lymphoma↑ Fibrinogen, D-dimer, prothrombin < fragment 1 + 2, FVIII, and VWF in most patients. Fibrinogen increased if IL-6 was high, and a correlation was observed between IL-6 and D-dimer.OGT is significantly upregulated in AML tissues compared with normal tissues. The high level of OGT expression is significantly related to poor overall survival in AML. Inhibition of OGT can inhibit AML cell proliferation and promote AML cell apoptosis.[191]Abbreviations: VWF, von Willebrand factor; TF, tissue factor; VEGF, vascular endothelial growth factor; AFP, α-Fetoprotein; MTA1, chromatin modifier metastasis-associated protein 1; hs-CRP, high-sensitivity C-reactive protein; CRP, C-reactive protein; MLR, monocyte to lymphocyte ratio; NLR, neutrophil to lymphocyte ratio; PLR, platelet to lymphocyte ratio; IL, interleukine; OGT, O-GlcNAc transferase. RhoA, Ras homolog family member A; ROCK, Rho-associated protein kinase; MLC, myosin light chain; HCC, hepatocellular carcinoma; LPLT, percentage of large platelets; IGF-1, insulin-like growth factor I; HPV E6, high-risk human papillomavirus oncogene, CRC, colorectal cancer; EZH2, histone methyltransferase enhancer of zeste homolog; Sox2, SRY-box transcription factor 2; ERK, extracellular signal-regulated kinase; ER, endoplasmic reticulum; RTN2, reticulon 2; RAB10, Ras-related protein; NAFLD, non-alcoholic fatty liver; JNK, c-Jun N-terminal kinase; AP-1, activator protein-1; NF-κB, nuclear factor-kappaB; AML, acute myeloid leukemia; ↑ increase; ↓ decreases.


In addition to coagulation or fibrinolysis abnormalities, cancer patients exhibit increased platelet responses. All of these, along with platelet alterations, are high-risk factors for thrombosis in cancer patients, as cancer cells can activate platelets and stimulate aggregation through different mechanisms [7,30], Figure 6; tumor cell-induced platelet aggregation (TCIPA) has been correlated with increased metastatic potential [192]. Tumor cells induce platelet aggregation by secreting thrombin, adenosine diphosphate (ADP) [193], and TF. Thrombin activates coagulation factors V, VIII, and XIII and protease-activated receptors (PARs) [194]. ADP activates platelets via the P2Y1 and P2Y12 receptors, causing the platelets to release more ADP from their dense granules and activate more platelets [195]. TF is the main activator of the coagulation cascade when it interacts with factor VIIa [196,197]. Platelets activated by cancer cells can regulate hematopoietic and immune cell migration to the tumor site, which may also contribute to cancer metastasis and progression by stimulating deep venous thrombosis [7].

On the other hand, platelets release proinflammatory cytokines that recruit and activate leukocytes [198] and growth factors that induce tumor growth and angiogenesis [199]; they also release P-selectin, which favors the formation of neutrophil extracellular traps (NETs) [200], while the release of Toll-like receptor 4 (TLR4) triggers NETosis in activated neutrophils [201]. NETs are structures derived from the chromatin and granular content of neutrophils [202], and they play a key role in defense by trapping and killing microorganisms [196]. Some studies suggest that they may be involved in tumor progression, metastasis, and cancer-associated thrombosis. NETs are procoagulant factors because they promote fibrin deposition, recruit red blood cells, and improve platelet activation [203]; in cancer patients, they mask cancer [151].

## 5. Applications of O-GlcNAcylation in Cancer

The metabolism of cancer cells is influenced by the availability of nutrients in the TME. This is detected by a mechanism that senses nutrients, such as O-GlcNAcylation, mainly by the hexosamine biosynthetic pathway [32]. The regulation of glycolysis and lipid metabolism by O-GlcNAc and OGT has been described. For example, in liver cancer, the depletion of PGK1 dramatically inhibited cancer cell glycolysis, proliferation, and tumorigenesis [204]. In colon cancer, the blocking of O-GlcNAcylation of PGK1 decreases cell proliferation and suppresses glycolysis [46]. The blocking PKM2 O-GlcNAcylation attenuated tumor growth in breast cancer and cervical cancer [205]. In breast cancer, the reduced O-GlcNAcylation leads to an increase in phosphorylated AMPK pathway and decreases cell proliferation via SREBP-1 regulation, and OGT suppression reduces levels of apoptotic marker cleaved PARP and decreases cell proliferation [206]. Other reports have shown that altered levels of O-GlcNAc and OGT are associated with the promotion of resistance to anti-cancer therapeutic agents [207]. Utilizing this knowledge could help in devising strategies to enhance the effectiveness of cancer treatments (Table 4). Furthermore, creating powerful and specific inhibitors of OGT or OGA may hold promise in treating diseases marked by abnormal O-GlcNAcylation [208].

## 6. Conclusions

The state of thrombosis or hypercoagulability associated with cancer is rarely considered, but it can lead to the patient´s death. Therefore, it is very important to consider that the increase in O-GlcNAc is involved in the reprogramming of metabolism in cancer cells as well as in the alteration of various pathways related to inflammation and hemostasis. For this reason, a better understanding of the O-GlcNacylation mechanisms could be considered as a potential for improved prognostic or clinical prediction tools in the health care of cancer patients, as well as helping facilitate the generation of therapeutic strategies that allow the prevention of complications. In conclusion, it is necessary to approach the diagnosis and treatment of cancer as a network that integrates multiple signaling pathways such as PI3K/Akt/c-Myc, the nuclear factor kappa B, and PI3K/AKT/mTOR, and to explore potential drug combinations that could regulate this signaling network.

## Figures and Tables

**Figure 1 ijms-25-09896-f001:**
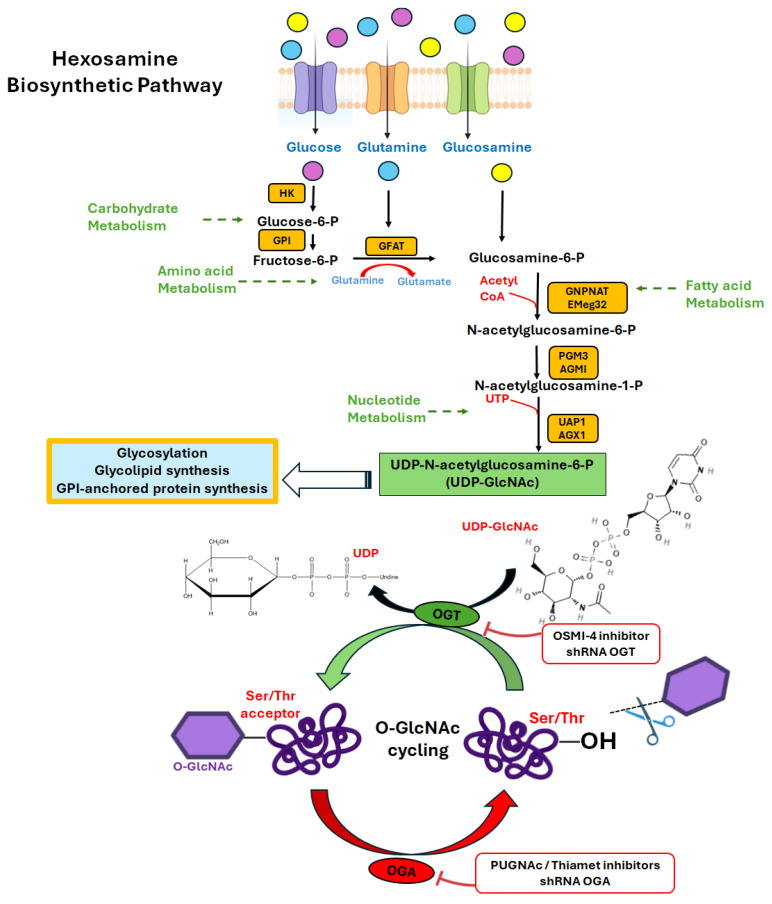
Modulating O-GlcNAcylation and some inhibitors. Abbreviations: HK, hexokinase; GPI, glucose-6-phosphate isomerase; GFAT, glutamine fructose-6-phosphate amidotransferase; GNPNAT/Emeg32, glucosamine–phosphate N-acetyltransferase; CoA, coenzyme A; GlcNAc, N-acetylglucosamine; PGM3, phosphoglucomutase 3; AGM1, phosphoacetylglucosamine mutase 1; UTP, uridine triphosphate; UDP, uridine diphosphate; UAP1, UDP-N-acetylglucosamine pyrophosphorylase 1; UDP-GlcNAc, UDP-N-acetylglucosamine; AGX1, UDP-N-acetylgalactosamine pyrophosphorylase; OGT, O-GlcNAc transferase; OGA, O-GlcNAcase; Ser, serine; Thr, threonine.

**Figure 2 ijms-25-09896-f002:**
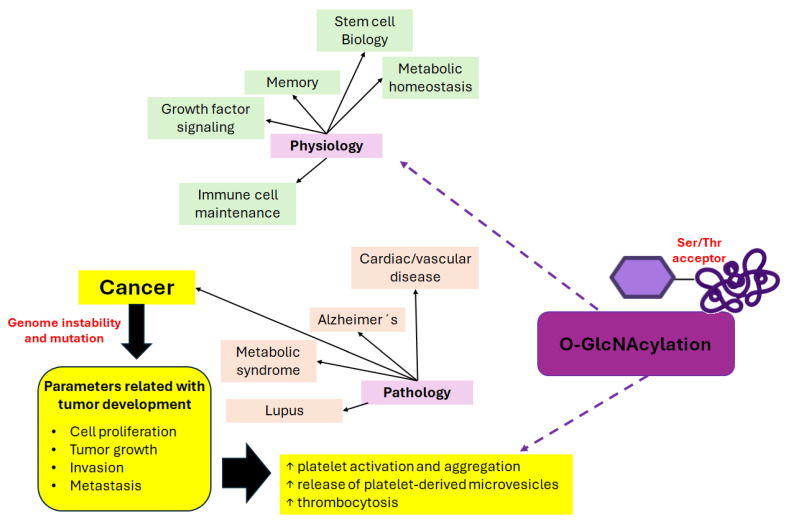
O-GlcNAcylation of proteins affects biological and pathological homeostasis.

**Figure 3 ijms-25-09896-f003:**
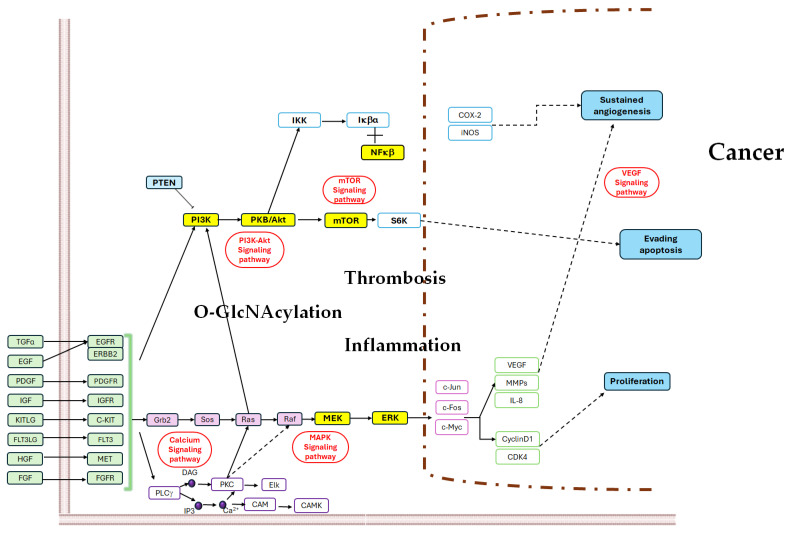
Pathways associated with O-GlcNAcylation, thrombosis, inflammation, and cancer. Abbreviations: TGFα, transforming growth factor alpha; EGF, epidermal growth factor; PDGF, platelet-derived growth factor subunit A; IGF, insulin-like growth factor; KITLG, KIT ligand; FLT3LG, fms-related tyrosine kinase 3 ligand; HGF, hepatocyte growth factor; FGF, fibroblast growth factor 1; EGFR, epidermal growth factor receptor; ERBB2, receptor tyrosine protein kinase erbB-2; PDGFR, platelet-derived growth factor receptor alpha; IGFR, insulin-like growth factor 1 receptor; C-KIT, proto-oncogene tyrosine protein kinase Kit; FLT3, fms-related tyrosine kinase 3; MET, proto-oncogene tyrosine protein kinase Met; FGFR, fibroblast growth factor receptor 1; Grb2, growth factor receptor-bound protein 2; Sos, son of sevenless; Ras, GTPase Ras; Raf, A-Raf proto-oncogene serine/threonine protein kinase; MEK or MAP2K1, mitogen-activated protein kinase kinase 1; ERK, mitogen-activated protein kinase 1/3; c-Myc, Myc proto-oncogene protein; c-Fos, proto-oncogene protein c-fos; c-Jun, transcription factor AP-1; VEGF, vascular endothelial growth factor; MMPs, matrix metalloproteinases (interstitial collagenase); IL-8, interleukin 8; CyclinD1, G1/S-specific cyclin-D1; CDK4, cyclin-dependent kinase 4; PLCγ, phosphatidylinositol phospholipase C, gamma-1; DAG, diacylglycerol; IP3, inositol 1,4,5-trisphosphate; Ca^2+^, calcium 2+; PKC, classical protein kinase C; CAM, calmodulin; CAMK, calcium/calmodulin-dependent protein kinase (CaM kinase) II; Elk, ETS domain-containing protein Elk-1; PTEN, phosphatidylinositol-3,4,5-trisphosphate 3-phosphatase and dual-specificity protein phosphatase PTEN; PI3K, phosphatidylinositol-4,5-bisphosphate 3-kinase catalytic subunit alpha/beta/delta; Akt, RAC serine/threonine protein kinase; mTOR, serine/threonine protein kinase mTOR; S6K, ribosomal protein S6 kinase beta; IKK, inhibitor of nuclear factor kappa-B kinase; Iκβα, NF-kappa-B inhibitor alpha; NFκβ, nuclear factor NF-kappa-B p105 subunit; COX-2, prostaglandin endoperoxide synthase 2; iNOS, nitric oxide synthase, inducible.

**Figure 4 ijms-25-09896-f004:**
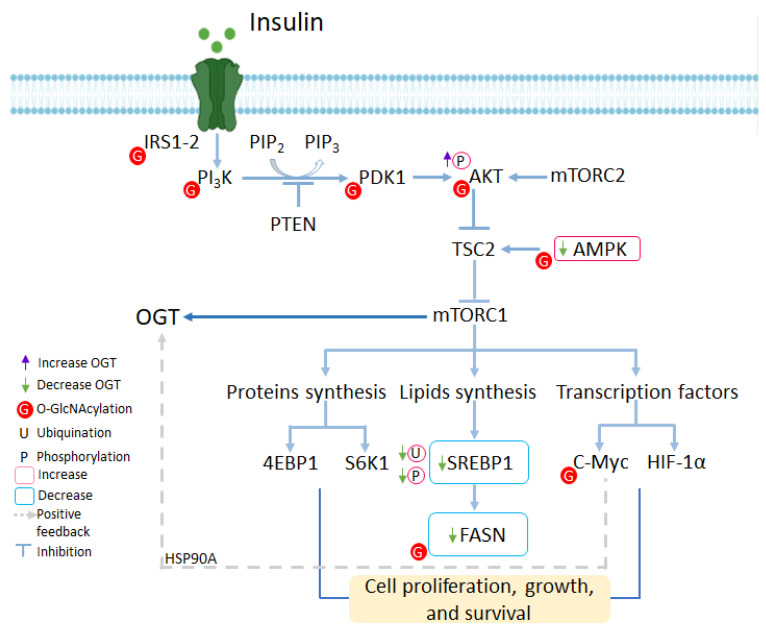
PI3K/Akt/mTOR signaling pathway by O-GlcNAcylation. Abbreviations: IRS1-2, insulin receptor substrates 1 and 2; PI_3_K, phosphatidylinositol 3-kinase; PTEN, phosphatidylinosi-tol-3,4,5-trisphosphate 3-phosphatase; PIP_2_, phosphatidylinositol bisphosphate; PIP_3_, phosphatidylinositol triphosphate; PDK1, 3-phosphoinositide-dependent protein kinase 1; Akt, Ser/Thr kinase protein kinase B; mTORC2, mechanistic target of rapamycin complex 1; AMPK, adenosine monophosphate-activated protein kinase; TSC2, tuberous sclerosis complex 2; mTORC1, mechanistic target of rapamycin complex 1; 4EBP1, eukaryotic translation initiation factor 4E-binding protein; S6K1, ribosomal protein S6 kinase 1; HIF-1α, hypoxia-inducible factor 1 alpha subunit; SREBP1, sterol regulatory element binding protein 1; FASN, fatty acid synthase; HSP90A, heat shock protein 90 alpha.

**Figure 5 ijms-25-09896-f005:**
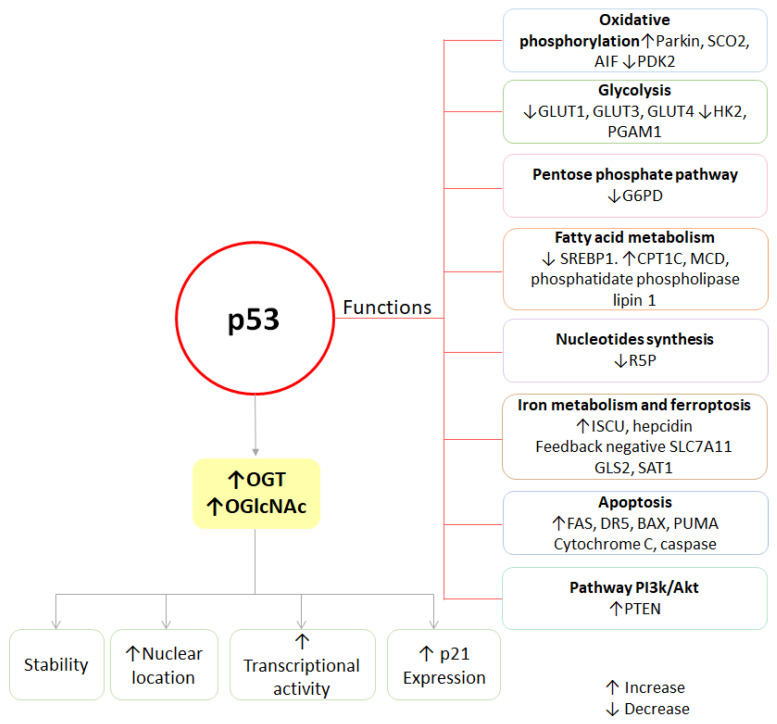
O-GlcNAcylation influences p53 functions. Abbreviations: GLUT: glucose transporters; HK2: hexokinase 2; PGAM1: phosphoglycerate mutase 1; CPT1C: carnitine palmitoyltransferase; SREBP1: sterol regulatory element-binding protein 1; R5P: ribose-5-phosphate; PTEN: phosphatidylinositol-3,4,5-trisphosphate 3-phosphatase; FAS: Fas receptor; DR5: death receptor 5; ISCU: iron–sulfur cluster assembly enzyme; SLC7A11: solute carrier family 7, member 11; GLS2: glutaminase 2; SAT1: spermidine/spermine N1-acetyltransferase 1; SCO2: synthesis of cytochrome c oxidase 2; AIF: mitochondrial protein apoptosis-inducing factor; PDK2: pyruvate dehydrogenase kinase 2; G6PD: glucose-6-phosphate dehydrogenase; MCD: malonyl-CoA decarboxylase; BAX: Bcl-2-associated X protein; PUMA: P53 up-regulated modulator of apoptosis.

**Figure 6 ijms-25-09896-f006:**
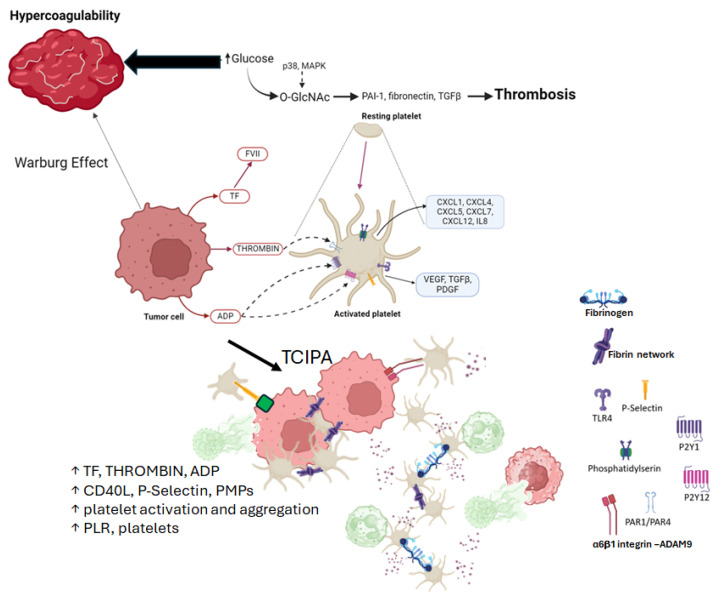
O-GlcNAcylation, a crosstalk between hemostasis, inflammation, and cancer. Abbreviations: ADP: adenosine diphosphate; VEGF: vascular endothelial growth factor; PDGF: platelet-derived growth factor; TGFβ: transforming growth factor β; CXCL: C-X-C motif chemokine 13; MAPK: mitogen-activated protein kinases; PAI-1: plasminogen activator inhibitor 1; TF: tissue factor; PMPs: platelet-derived microparticles; CD40L: CD40 ligand; PLR: index platelet to lymphocyte ratio.

**Table 1 ijms-25-09896-t001:** Infectious agents, types of cancer, and virulence factors associated with chronic inflammation.

Infectious Agents Associated with Cancer	Cancer Types with Increased Protein O-GlcNAcylation [106]	Virulence Factors	Action through Modulation of Immune–Inflammatory Pathways	References
*Helicobacter pylori*	Gastric, primary gastric non-Hodgkin’s lymphoma	Cytotoxin-associated gene A (CagA), vacuolating cytotoxin A (VacA), cytotoxin-associated gene pathogenicity island (cagPAI), neutrophil-activating protein (NAP)	↑ Smad7 ↑ROS, RNS ↑ IL-12, IL-17, IL-21 ↑ IFN-γ, TNF-α ↑ MMP-2, MMP-9 ↑ NF-κB activity	[107,108,109]
Hepatitis B virus (HBV)	Hepatocellular carcinoma	HBV surface antigen (HBsAg), HBV x protein (HBx)	↑ NF-κB, MAPKs, JAK/STAT ↑ IFN-α, TNF-α, IL-15 ↑ IL-1β, IL-18 ↑ TAK1 activity ↑ IL-34 by activation STAT3 ↑ TGF-β, IL-10 ↑ miR-1269b by NF-κB dependent	[110,111,112]
Human papillomavirus (HPV)	Skin, cervical, penile, vulvovaginal, anal	E proteins/genes	↑ COX-1/COX-2, MMP, TNF-α ↑ ROS, RNS ↓ TLR 9, IFN-γ ↑ MDSCs ↓ cytotoxic T lymphocytes ↑ IL-10, TGF-β EGFR/NF-κB activation	[113,114,115]
*Propionibacterium acnes*	Prostate, gastric	Adhesive Flp pili encoded by a gene locus for tight adherence (tad)	↑ M2 polarization of macrophages by TLR4/PI3K/Akt signaling ↑ IL-4, IL-10 ↑ IL-6, IL-8 ↑ JAK/STAT, IL-17	[116,117,118]
*Schistosoma haematobium*	Bladder squamous cell carcinoma	Soluble egg antigens, IPSE/alpha-1 protein	↑ STAT4 (Th1), GATA3 (Th2), FOXP3 (T regulatory), CD8 (T cytotoxic) ↑ IL-10, TGF-β	[119,120,121,122]
*Klebsiella pneumoniae*	Colorectal	Fimbriae virulence factors (mrkD, mrkF, mrkC)	↑ JAK/STAT NF-κB activation via the Akt/IκB kinase ↑ COX-2, ROS	[123,124,125]
*Escherichia coli*		Fimbriae virulence factors (FimE, FimB, FimG, erpC, FimA)		
*Yersinia pestis*		Yersiniabactin (ybtE, fyuA, Ybtp, Irp1, Irp2)		

Abbreviations: Smad7: suppressor of mothers against decapentaplegic 7; ROS: reactive oxygen species; RNS: reactive nitrogen species; IL: interleukin; MMP: matrix metalloproteinase; NF-κB: nuclear factor kappa-light-chain-enhancer of activated B cells; JAK: Janus kinase; STAT: signal transducer and activator of transcription; TAK1: factor-β-activated kinase 1; TGF-β: transforming growth factor-β; TLR: Toll-like receptor; MDSCs: immunosuppressive myeloid-derived suppressor cells; EGFR: epidermal growth factor receptor; ↑ increase; ↓ decreases.

**Table 4 ijms-25-09896-t004:** Inhibitors of O-GlcNAcylation and effect on platelets activity.

Drugs/Molecule	Mechanism of Action and Pathway	Effect on Platelets Activity	References
OSMI-1	Inhibition of OGT activates tumor-suppressor gene expression in tamoxifen-resistant breast cancer cells. Enhances TRAIL-induced apoptosis through ER stress and NF-κB signaling in colon cancer cells. Also, decreased expression of NKG2D and NKG2A receptors; cytokines including TNF-α and IFN-γ; cytotoxic mediators perforin, granzyme B, soluble Fas ligand, and granulysin, resulting in reduced cytotoxic function of NK cells.	Inhibition of OGT and O-GlcNAcylation induces megakaryocyte differentiation and platelet production through c-Myc stabilization and integrin perturbation. Absolute platelet counts revealed an increase in the number of CD41a+ platelet-like particles (PLPs) and CD42b+ PLPs, thereby validating the platelet-promoting role of OGT inhibition by OSMI-1.	[69,209,210,211,212,213]
ST 045849	Depletes intracellular alanine and decreases glucose consumption by cancer cells; reduces proliferation and viability of prostate cancer cells.		[214,215]
L01	Specific inhibitor of OGT and has low toxicity in cellular and zebrafish models. Inhibit cell proliferation by reducing the O-GlcNAcylation of proteins related to proliferation. Also, it could bind to N557, located near the UDP-binding pocket of OGT, and might contribute to the specificity of L01.	Inhibition of OGT by small molecule inhibitors facilitates differentiation of hematopoietic stem and progenitor cells into megakaryocytes and stimulates platelet production, suggesting that reduced O-GlcNAcylation promotes megakaryopoiesis and thrombopoiesis.	[216,217]
Benzyl-2-acetamido-2-deoxy-α-d-galactopyranoside (BADGP)	BADGP acts as a GalNAc-α-1-O-serine/threonine mimic and, thus, as a competitive inhibitor of O-glycan chain extension by blocking the β1,3-galactosyltransferase involved in O-glycosylation elongation. Also, reduces O-GlcNAc level, but lacks specificity for OGT.		[218,219,220]
Streptozotocin (STZ)	STZ is primarily used to treat neuroendocrine tumors. STZ is incorporated into pancreatic β-cells through glucose transporter type 2, thereby promoting DNA damage, reactive oxygen species production, mitochondrial dysfunction, and subsequent apoptosis. STZ inhibits O-GlcNAcase via the production of a transition state analog.	STZ is known to trigger a slow activation of coagulation, resulting in the consumption of circulating fibrinogen and loss of platelet aggregation.	[221,222,223,224]
Alloxan	Alloxan is an inhibitor of O-GlcNAc-selective N-acetyl-beta-d-glucosaminidase, with inhibition corresponding to an altered tryptic digest pattern of N-terminal active site peptides.	The clotting time of whole blood and recalcified plasma was slightly prolonged. Tissue plasmin inhibition showed no significant difference.	[225,226,227]
Thiamet G	Thiamet-G binds to OGA in competition with 4-methylumbelliferyl N-acetyl-β-d-glucosaminide dehydrate, an analogue of O-GlcNAc UDP, thereby suppressing the activity of OGA; inhibition of OGA by thiamet-G decreased the phosphorylation of microtubule-associated protein Tau and caused alterations of microtubule network in cells. Also, thiamet-G-mediated inhibition of O-GlcNAcase sensitizes human leukemia cells to microtubule-stabilizing agent paclitaxel.		[228]

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
