# Peer review of "O-GlcNAcylation: Crosstalk between Hemostasis, Inflammation, and Cancer"

_ijms, 2024, doi:10.3390/ijms25189896_

Round 1
Reviewer 1 Report (Previous Reviewer 2)
Comments and Suggestions for Authors
The resubmitted manuscript has been improved much. The authors should nevertheless address the following points:
1. In the introduction, the authors state that ‘Pathways associated with O-GlcNAcylation, thrombosis, inflammation and cancer are the PI3K/Akt/c-Myc pathway [8], the nuclear factor-kappa B (NF-κB) signaling pathway [9,10] and PI3K/AKT/mTOR pathway [11]’. Are these pathways the only pathways related to O-GlcNAcylation, thrombosis, inflammation and cancer? Are they associated with O-GlcNAcylation, thrombosis, inflammation and cancer at the same time?
2. It would be appreciated if a brief summary could be provided in the manuscript covering the contents in Section 2 and Section 3.
3. I would recommend the authors to include some applications of O-GlcNAcylation in health care of cancer patients, or their own opinion and perspective.
Comments on the Quality of English LanguageMy recommendation would be for the authors to carefully review and revise the manuscript to correct any grammatical errors and typos.
Author Response
The resubmitted manuscript has been improved much. The authors should nevertheless address the following points:
Q1. In the introduction, the authors state that ‘Pathways associated with O-GlcNAcylation, thrombosis, inflammation and cancer are the PI3K/Akt/c-Myc pathway [8], the nuclear factor-kappa B (NF-κB) signaling pathway [9,10] and PI3K/AKT/mTOR pathway [11]’. Are these pathways the only pathways related to O-GlcNAcylation, thrombosis, inflammation and cancer? Are they associated with O-GlcNAcylation, thrombosis, inflammation and cancer at the same time?
Answer. We appreciate your comments to improve the work. They are not the only ones, but they are the most studied pathways associated with O-GlcNAcylation, thrombosis, inflammation and cancer. We added a web link where the cancer pathways are shown and added a figure where the pathways of O-GlcNAcylation, thrombosis, inflammation and cancer coincide.
Q2. It would be appreciated if a brief summary could be provided in the manuscript covering the contents in Section 2 and Section 3.
Answer. We appreciate your comments to improve the work. We have now modified section two and 3.
Q3. I would recommend the authors to include some applications of O-GlcNAcylation in health care of cancer patients, or their own opinion and perspective.
Answer. We appreciate your comments to improve the work. We now add some applications of O-GlcNAcylation in health care of cancer patients.
Comments on the Quality of English Language:
My recommendation would be for the authors to carefully review and revise the manuscript to correct any grammatical errors and typos.
Answer. We appreciate your comments to improve the work. We check for spelling errors.

Reviewer 2 Report (New Reviewer)
Comments and Suggestions for Authors
In this manuscript, the authors presented a comprehensive review of the relationship between O-GlcNAcylation and cancer, tumor progression, inflammation, and hemostasis. The authors first introduced the O-GlcNAcylation, its reader and eraser proteins, and the corresponding signaling pathways, then discussed the crosstalk between O-GlcNAcylation and hemostasis, inflammation, and cancer. Overall, this manuscript makes a valuable contribution to the field of cancer-related O-GlcNAcylation, providing a thorough and insightful review of current knowledge. Below are some suggestions for enhancement.
Introduction: Including a scheme that shows the structure of O-GlcNAcylation and the installation and removal of O-GlcNAcylation by O-GlcNAc transferase (OGT) and O-GlcNAcase (OGA) would make the manuscript accessible to a broader audience, including those not in relevant fields. The authors should also include the purpose and scope of the review in the introduction.
O-GlcNAcylation, cancer and hemostasis: It is acceptable to include part of the O-GlcNAcylation introduction, such as O-GlcNAcylation alterations in diseases beyond cancer, in this session. However, it would be better if the authors could clearly introduce O-GlcNAcylation, cancer-related hemostasis, and cancer-related inflammation regulation one by one and move these introductions to the introduction session. Additionally, there is extensive content regarding the signaling pathways regulated by O-GlcNAcylation. The authors should make this part concise and clear. Detailed functional explanations of each protein involved in the signaling pathways should be shortened and focused on the interrelationship between them and O-GlcNAcylation.
Conclusions: The authors stated the importance of thrombosis and hypercoagulability in cancer and the critical role of O-GlcNAc in metabolic reprogramming. It would be beneficial if the authors could provide more insights into future directions for the investigation of O-GlcNAc in cancer, such as identifying key missing parts of current research, explaining how O-GlcNAc can serve as a prognostic or clinical prediction tool.
Minor points:
Given that the review discussed the relationship between O-GlcNAc and other cancer-related cellular processes, the authors should change the title of the manuscript to avoid misunderstanding.
Line 26: I would recommend changing “We aim to show the relationship between O-GlcNAcylation, hemostasis, inflammation, and cancer…” to “We aim to show the relationship between O-GlcNAcylation and hemostasis, inflammation, cancer…”
Line 43: The full name of O-GlcNAcylation should be listed when it first appears, which is in Line 33.
Figure 1 Legend, “Homeostasis” should be changed to “Hemostasis”
The first letter of each word in subtitles should be capitalized. Additionally, Title “2.1…” is missing.
Author Response
Reviewer 2
Comments and Suggestions for Authors
In this manuscript, the authors presented a comprehensive review of the relationship between O-GlcNAcylation and cancer, tumor progression, inflammation, and hemostasis. The authors first introduced the O-GlcNAcylation, its reader and eraser proteins, and the corresponding signaling pathways, then discussed the crosstalk between O-GlcNAcylation and hemostasis, inflammation, and cancer. Overall, this manuscript makes a valuable contribution to the field of cancer-related O-GlcNAcylation, providing a thorough and insightful review of current knowledge. Below are some suggestions for enhancement.
Comment: Introduction: Including a scheme that shows the structure of O-GlcNAcylation and the installation and removal of O-GlcNAcylation by O-GlcNAc transferase (OGT) and O-GlcNAcase (OGA) would make the manuscript accessible to a broader audience, including those not in relevant fields. The authors should also include the purpose and scope of the review in the introduction.
Answer: We appreciate your comments to improve the work, now we add a scheme that shows the structure of O-GlcNAcylation and the purpose and scope of the review in the Introduction.
Comment: O-GlcNAcylation, cancer and hemostasis: It is acceptable to include part of the O-GlcNAcylation introduction, such as O-GlcNAcylation alterations in diseases beyond cancer, in this session. However, it would be better if the authors could clearly introduce O-GlcNAcylation, cancer-related hemostasis, and cancer-related inflammation regulation one by one and move these introductions to the introduction session. Additionally, there is extensive content regarding the signaling pathways regulated by O-GlcNAcylation. The authors should make this part concise and clear. Detailed functional explanations of each protein involved in the signaling pathways should be shortened and focused on the interrelationship between them and O-GlcNAcylation.
Answer: We appreciate your comments to improve the work, we added an image to show the relationship.
Conclusions: The authors stated the importance of thrombosis and hypercoagulability in cancer and the critical role of O-GlcNAc in metabolic reprogramming. It would be beneficial if the authors could provide more insights into future directions for the investigation of O-GlcNAc in cancer, such as identifying key missing parts of current research, explaining how O-GlcNAc can serve as a prognostic or clinical prediction tool.
Answer: We appreciate your comments to improve the work, we improve the conclusions according to the comments.
Minor points:
Given that the review discussed the relationship between O-GlcNAc and other cancer-related cellular processes, the authors should change the title of the manuscript to avoid misunderstanding.
Answer: We appreciate your comments to improve the work, we consider that the title of the work may be adequate.
Line 26: I would recommend changing “We aim to show the relationship between O-GlcNAcylation, hemostasis, inflammation, and cancer…” to “We aim to show the relationship between O-GlcNAcylation and hemostasis, inflammation, cancer…”
Answer: We appreciate your comments to improve the work, we changed the wording of the objective.
Line 43: The full name of O-GlcNAcylation should be listed when it first appears, which is in Line 33.
Answer: We appreciate your comments to improve the work, we modified the line (now line 36).
Figure 1 Legend, “Homeostasis” should be changed to “Hemostasis”
Answer: We appreciate your comments to improve the work, but, it is correct homeostasis because the figure refers to normal and pathological O-GlcNAcylation of proteins
The first letter of each word in subtitles should be capitalized. Additionally, Title “2.1…” is missing.
Answer. We appreciate your comments to improve the work. We have now modified section.

Round 2
Reviewer 2 Report (New Reviewer)
Comments and Suggestions for Authors
I appreciate the efforts the authors made to address the previous comments and enhance the overall quality of the study. The revised manuscript has successfully addressed the previous concerns, and the study now presents a more compelling and well-substantiated body of review. The manuscript’s contributions to our understanding of O-GlcNAcylation and its roles in hemostasis, inflammation and cancer. I recommend the manuscript for publication and commend the authors for their diligent work and responsiveness to feedback.
This manuscript is a resubmission of an earlier submission. The following is a list of the peer review reports and author responses from that submission.
Round 1
Reviewer 1 Report
Comments and Suggestions for Authors
The authors present a review regarding the importance of O-GlcNAcylation in hemostasis, inflammation and cancer development and progression. In principle this is an interesting topic, however, this paper does not meet expectations. It is very weakly structured with a lot of repetitions in different chapters. Several biological processes and many signalling pathways are described, but the connection to O-GlcNAcylation is not clearly given. The authors have a lot of knowledge about different cancers and processes in cancer cells, but the connection to the topic of the review is weak.
In detail:
The text in the “introduction” is quite similar to the one in chapter 2. This should be combined.
The molecular weights in chapter 2 are rather random. They do not fit together. The information from the references 6, 9 and 11 should be merged in a better way.
Figure 1 is misleading. It looks like the production of UDP-GlcNAc has a direct effect on carbohydrate metabolism, glycogen synthesis, amino acid metabolism, fatty acid metabolism and nucleotide metabolism. Of course, there is a connection to these pathways, as all metabolic pathways are somehow related, but there is certainly no direct influence.
Figure 2 gives a nice overview on insulin signalling, but the connection to O-GlcNAcylation is missing.
Figure 3 summarises p53 putative functions. Where is the O-GlcNAc?
Table 1 gives associates cancer with inflammation. Where is the O-GlcNAc?
Table 2: O-GlcNAc glycosylation sites of transcription factors are given. These factors have important functions in up/down regulations of signal pathways, but is their action regulated by O-GlcNAcylation? Or just by (over)expression of the factor under different conditions?
Chapter 2.3 with table 3 is the only fitting chapter in this paper. Here the connection is clearly given.
My suggestion:
Write a review around chapter 2.3 and forget the rest, or write a review about cancer and remove “O-GlcNAcylation” from the title. In the current state, the paper is not publishable.
Furthermore, please improve the language. It is VERY difficult to read and several Spanish words are included (factor de crecimiento transformante, gástrico, ..)
Comments on the Quality of English LanguageVERY difficult to read. Some spanish left overs; wrong prepositons, missing verbs, "these" and "those" refer to nothing, wrong word order, ...
Examples just from the 1st page: O-GlcNAcylation is a process highly dynamic and reversible post-translational modification of proteins.
or
In recent years, has been shown to role of O-GlcNAcylation ....
Author Response
Reply to Reviewer 1
The authors present a review regarding the importance of O-GlcNAcylation in hemostasis,
inflammation and cancer development and progression. In principle this is an interesting topic,
however, this paper does not meet expectations. It is very weakly structured with a lot of repetitions
in different chapters. Several biological processes and many signalling pathways are described, but the
connection to O-GlcNAcylation is not clearly given. The authors have a lot of knowledge about
different cancers and processes in cancer cells, but the connection to the topic of the review is weak.
In detail:
Comment 1: The text in the “introduction” is quite similar to the one in chapter 2. This should be
combined.
Reply 1: We appreciate your comments, now we have combined both sections.
Comment 2: The molecular weights in chapter 2 are rather random. They do not fit together. The
information from the references 6, 9 and 11 should be merged in a better way.
Reply 2: We appreciate your comments. We have corrected this information and checked references
according to context.
Comment 3: Figure 1 is misleading. It looks like the production of UDP-GlcNAc has a direct effect on
carbohydrate metabolism, glycogen synthesis, amino acid metabolism, fatty acid metabolism and
nucleotide metabolism. Of course, there is a connection to these pathways, as all metabolic pathways
are somehow related, but there is certainly no direct influence.
Reply 3: We appreciate your comments and have now improved the figure 1.
Comment 4: Figure 2 gives a nice overview on insulin signalling, but the connection to OGlcNAcylation is missing.
Reply 4: We appreciate your comments, we have now added the connection with O-GlcNAcylation.
Comment 5: Figure 3 summarises p53 putative functions. Where is the O-GlcNAc?
Reply 5: We appreciate your comments, we have now added the connection with O-GlcNAcylation.
Comment 6: Table 1 gives associates cancer with inflammation. Where is the O-GlcNAc?
Reply 6: We appreciate your comments to improve the work, now we add the association of cancer,
inflammation with O-GlcNAcylation and, action through the modulation of immune-inflammatory
pathways.
Comment 7: Table 2: O-GlcNAc glycosylation sites of transcription factors are given. These factors have
important functions in up/down regulations of signal pathways, but is their action regulated by OGlcNAcylation? Or just by (over)expression of the factor under different conditions?
Reply 7: We appreciate your comments to improve the work, now we add a column in the table and
associate the overexpression of O-GlcNacylation in cancer.
Comment 8: Chapter 2.3 with table 3 is the only fitting chapter in this paper. Here the connection is
clearly given.
Reply 8: We appreciate your comment, and we also improved figure 3 of this section.
Comment 9:
My suggestion:
Write a review around chapter 2.3 and forget the rest, or write a review about cancer and remove “OGlcNAcylation” from the title. In the current state, the paper is not publishable.
Furthermore, please improve the language. It is VERY difficult to read and several Spanish words are
included (factor de crecimiento transformante, gástrico, ..)
Reply 9: We appreciate your time in reviewing the work and your comments to improve it. We also
apologize for those errors. We have now improved the work.
Comment 10:
Comments on the Quality of English Language
VERY difficult to read. Some spanish left overs; wrong prepositons, missing verbs, "these" and "those"
refer to nothing, wrong word order, ...
Examples just from the 1st page: O-GlcNAcylation is a process highly dynamic and reversible posttranslational modification of proteins.
or
In recent years, has been shown to role of O-GlcNAcylation ....
Reply 10: We made modifications to the manuscript, and the changes suggested by the reviewers in
red text.
Reviewer 2 Report
Comments and Suggestions for Authors
In this review, the authors tried to summarize the relationship between O-GlcNAcylation and hemostasis and immunity in cancer. Proteins and signaling pathways with various functions in cancer were discussed in details, including PGK1, ENO1, G6PD, PI3K/Akt/mTOR pathway etc. The authors concluded that hypercoagulability induced by cancer could increase O-GlcNAcylation level which is related to inflammation, hemostasis, and immunity in cancer. However, the alteration in O-GlcNAcylation involved lacks clear demonstration. And some other concerns are as follows:
1. The abstract provides little information about the hypothesis or conclusion of this manuscript.
2. The significance of the topic is not clearly stated in the introduction section.
3. In Section 2.1 ‘O-GlcNAcylation and cancer’, the authors described the functions of PGK1 and ENO1. But what are the relationships between the two proteins (or their O-GlcNAcylation levels) and cancer? Similarly, according to the authors, PI3K/Akt/mTOR pathway is involved in growth and control of glucose metabolism, as well as in various cancers. However, the links between O-GlcNAcylation to cancers related to this pathway is not clearly described. In addition, at the end of this section, the authors introduced p53 as a tumor suppressor and described its function in suppressing glycolysis, but failed to explicitly explain the alteration of O-GlcNAcylation. Therefore, the conclusion of this section on O-GlcNAcylation appears disjointed.
4. The authors concluded in Section 3 that hypercoagulability induced by cancer could increase O-GlcNAcylation levels which is related to inflammation, hemostasis and immunity in cancer. However, it seems hard to draw this conclusion based on the proteins, their functions, and their O-GlcNAcylation levels described in Section 2.
Comments on the Quality of English Language
Some of the sentences are hard to understand, and there are typos to correct. For examples: L35-36, L39-41, L56-58, L65-67, L87-88, L118-120, L126-128, L133-136.
Author Response
Reply to Reviewer 2
In this review, the authors tried to summarize the relationship between O-GlcNAcylation and
hemostasis and immunity in cancer. Proteins and signaling pathways with various functions in cancer
were discussed in details, including PGK1, ENO1, G6PD, PI3K/Akt/mTOR pathway etc. The authors
concluded that hypercoagulability induced by cancer could increase O-GlcNAcylation level which is
related to inflammation, hemostasis, and immunity in cancer. However, the alteration in OGlcNAcylation involved lacks clear demonstration. And some other concerns are as follows:
Comment 1. The abstract provides little information about the hypothesis or conclusion of this
manuscript.
Reply 1: We appreciate your comments, now we improve the abstract.
Comment 2. The significance of the topic is not clearly stated in the introduction section.
Reply 2: We appreciate your comments, we have now improved the introduction.
Comment 3. In Section 2.1 ‘O-GlcNAcylation and cancer’, the authors described the functions of PGK1
and ENO1. But what are the relationships between the two proteins (or their O-GlcNAcylation levels)
and cancer? Similarly, according to the authors, PI3K/Akt/mTOR pathway is involved in growth and
control of glucose metabolism, as well as in various cancers. However, the links between OGlcNAcylation to cancers related to this pathway is not clearly described. In addition, at the end of this
section, the authors introduced p53 as a tumor suppressor and described its function in suppressing
glycolysis, but failed to explicitly explain the alteration of O-GlcNAcylation. Therefore, the conclusion
of this section on O-GlcNAcylation appears disjointed.
Reply 3: We appreciate your time in reviewing the work and your comments to improve it. We have
now improved the work.
Comment 4. The authors concluded in Section 3 that hypercoagulability induced by cancer could
increase O-GlcNAcylation levels which is related to inflammation, hemostasis and immunity in cancer.
However, it seems hard to draw this conclusion based on the proteins, their functions, and their OGlcNAcylation levels described in Section 2.
Reply 4. We appreciate your comments, now we improve the information.
Comment 5.
Comments on the Quality of English Language: Some of the sentences are hard to understand, and
there are typos to correct. For examples: L35-36, L39-41, L56-58, L65-67, L87-88, L118-120, L126-128,
L133-136.
Reply 5. We appreciate your comments, now we have revised the writing again and eliminated writing
errors.